# Phenolic Profiles of Red Wine Relate to Vascular Endothelial Benefits Mediated by SIRT1 and SIRT6

**DOI:** 10.3390/ijms22115677

**Published:** 2021-05-26

**Authors:** Nunzia D’Onofrio, Elisa Martino, Giuseppina Chianese, Francesca Coppola, Luigi Picariello, Luigi Moio, Maria Luisa Balestrieri, Angelita Gambuti, Martino Forino

**Affiliations:** 1Department of Precision Medicine, University of Campania Luigi Vanvitelli, Via L. De Crecchio 7, 80138 Naples, Italy; nunzia.donofrio@unicampania.it (N.D.); elisa.martino@unicampania.it (E.M.); 2Department of Pharmacy, University of Naples Federico II, Via D. Montesano, 49, 80131 Naples, Italy; g.chianese@unina.it; 3Department of Agricultural Sciences, Section of Vine and Wine Sciences, University of Naples Federico II, Viale Italia, 83100 Avellino, Italy; francesca.coppola2@unina.it (F.C.); luigi.picariello@unina.it (L.P.); luigi.moio@unina.it (L.M.); angelita.gambuti@unina.it (A.G.); forino@unina.it (M.F.)

**Keywords:** red wines, polyphenols, antioxidant, insulin resistance, hyperglycemia, sirtuins

## Abstract

Dietary phenolic compounds possess potent bioactivity against inflammatory pathways of chronic inflammatory conditions, such as type 2 diabetes. Here, the phenolic profile and bioactivity of Italian red wines Gaglioppo, Magliocco, and Nerello Mascalese were characterized. NMR, HPLC/UV-Vis and spectrophotometric characterization showed that Magliocco was the richest wine in monomeric anthocyanins (two-fold), catechins, and low molecular weight phenolics (LMWP). A positive correlation was observed between the polyphenolic content and antioxidant capacity (*p* < 0.05), with Magliocco displaying the highest antioxidant capacity (*p* < 0.01). In vitro evidence on the endothelial cell models of insulin resistance and hyperglycemia showed the ability of Magliocco to reduce reactive oxygen species (ROS) (*p* < 0.01) and cytokine release (*p* < 0.01) and to upregulate SIRT1 and SIRT6 (*p* < 0.01). On the whole, the results indicated that the quantitative and qualitative phenolic profiles of red wines influence their in vitro beneficial effects on oxidative and proinflammatory milieu in endothelial cells, showing a positive modulation of SIRT1 and SIRT6, both implied in vascular aging.

## 1. Introduction

Epidemiological studies advise that the moderate consumption of red wine reduces the incidence of age-related disease, including cardiovascular disease, type 2 diabetes, neurodegenerative diseases, and certain types of cancer [1]. As a component of the Mediterranean dietary pattern, the consumption of red wine is associated with improved age-related health due to its richness in polyphenols and other bioactive microconstituents [2,3,4,5]. Several studies have described the impact of polyphenols on age-related diseases, which are characterized by enhanced mitochondrial and peroxisomal dysfunction and by progressive altered redox homeostasis, cell senescence, and cell death, favoring heart and vessel injuries [6,7]. Red wine polyphenols are particularly attractive for their properties to target sirtuin 1 (SIRT1) and sirtuin 6 (SIRT6), two members of the sirtuin family of proteins mediating a large variety of pathophysiological events, including protection of the circulatory system function and prevention of endothelial dysfunction at the early stages of the pathogenesis of type 2 diabetes and atherosclerosis [6,7,8]. Natural polyphenols, first identified as sirtuin activators, compose a diversified class of molecules usually divided into two categories: flavonoids and nonflavonoids [9]. Over the past decades, because of their chemical complexities, studies on phenolic compounds have been essentially limited to low molecular mass molecules. Only recently, the ever-increasing efficiency of the analytical techniques has accelerated chemical and biological studies on natural polyphenols occurring in organic matrices as oligomeric or even larger polymeric molecules. This has been a significant step forward, especially for enological studies, as polymeric polyphenols are much more abundant than monomeric ones, both in grapes and wine [10]. The major red grape polyphenols are monomeric anthocyanins, together with proanthocyanidins (detected as monomers, oligomers, and polymers), even though, in lower amounts, other phenolics have been identified, such as phenolic acids, flavonols, flavanonols, flavones, and stilbenes. In red wines, in addition to grape polyphenols, there are further phenolic compounds formed during the winemaking process by either enzymatic or nonenzymatic reactions or even released from oak during the maturation process. As a result, the phenolic profile, especially in red wines, is quite complex and strongly dependent upon the grape cultivar and the production technology. Thus far, little is known about the chemistry of polymeric polyphenols and their actual compositions in wines. In this regard, the employment of different, yet complementary, techniques can be a successful strategy to overcome such a drawback. Recently, accelerated oxidation tests to simulate the oxidation of Magliocco, Gaglioppo, and Nerello Mascalese red wines during aging in regard to the phenolic composition and wine color demonstrated that the addition of hydrogen peroxide is a successful tool to predict the evolution of different phenolic compounds during air saturation-based treatments of wines [11].

In the present study, we investigated the relationship between the phenolic content and the bioactivity of red wines. To this end, three red wines were selected on the basis of the comparable pedoclimatic conditions under which grapes were grown and the same technological protocol adopted for their production. This was instrumental in ruling out any technological variables, thus allowing a straightforward correlation between the bioactivities of the wines and their specific content of healthy natural metabolites. The Italian red wines chosen were Magliocco, Gaglioppo, and Nerello Mascalese, all produced in the Calabria region (Italy). In particular, two main aspects were covered by this study: (a) chemical characterization and differentiation of the most abundant phenolics by means of an integrated approach of NMR, HPLC, and spectroscopic techniques and (b) the in vitro bioactivity in endothelial models of insulin resistance and hyperglycemia and potency as SIRT1 and SIRT6 activators.

## 2. Results

### 2.1. Chemical Characterization of Wines by NMR/ HPLC-UV/Vis and Spectrophotometric Analysis

The analysis of the base parameters of wines were determined by using official methods of analysis (Appendix A Appendix A). In order to identify the major phenolic compounds, both monomeric and polymeric, occurring in the analyzed wines, we resorted to three different analytical approaches: NMR, HPLC, and spectrophotometry. The choice of these three techniques was driven by the different, yet complementary, pieces of information they can provide. First, NMR constitutes a valid untargeted analysis to rapidly obtain a comprehensive qualitative and quantitative picture of the most abundant low molecular weight metabolites present in a given sample. The HPLC-based analysis, conducted according to the OIV official methods, is a robust tool to define the monomeric anthocyanins in red wine; finally, spectrophotometric measurements give crucial information on polymeric pigments that would be quite difficult to be analyzed otherwise. In addition to the compounds recently characterized in these red wines [11], the NMR-based analyses of Gaglioppo, Magliocco, and Nerello Mascalese brought to light different polyphenolic contents from both a qualitative and quantitative standpoint (Table 1).

The identity of each metabolite was ascertained either by comparing the detected chemical shifts with those reported in the literature or with those of authentic samples (Appendix A Appendix A). More specifically, besides anthocyanins, the major low molecular weight polyphenols detected in all of the three wines were catechin and epicatechin, belonging to the flavan-3-ol class, gallic acid, pyrogallol, tyrosol, and 2-phenylethanol. Additionally, ethyl caffeate was detected in Gaglioppo and Nerello Mascalese, while, in Magliocco, such a metabolite was either absent or occurring below the detection limit of NMR. All mentioned polyphenols were quantitated by means of Equation (1), apart from 2-phenylethanol because of its high volatility.

Further chemical analyses were conducted to better characterize the phenolic contents of the three wines under investigation. By employing the HPLC-based official OIV method, the contents of the monomeric anthocyanins in each wine were evaluated. Malvidin turned out to be the most abundant anthocyanin in the Magliocco wine, where it accounted for 86% of the total monomeric anthocyanins (Table 1), consistent with the previous data obtained on Magliocco [12] and in accordance with what has been reported for most red grape cultivars [13]. Malvidin usually occurs as malvidin-3-O-glucoside (Mv-3-glc), but it can also present either coumaroyl (Mv-3-cum) or acetyl (Mv-3-ace) moieties. In Magliocco, the malvidin composition expressed as mg/L was found to be the following: Mv-3-glc >> Mv-3-cum > Mv-3-ace. Such a composition is similar to that of several Italian grape varieties [13,14]. Even in Gaglioppo and Nerello, the most abundant anthocyanin was malvidin, with a relative abundance of 76% in Gaglioppo and 56% in Nerello, respectively. Again, the little amount of acylated anthocyanins we found in the Gaglioppo wine might be due to the significant percentage of Magliocco it contained. Finally, the spectrophotometric measures—instrumental in obtaining crucial information on polymeric phenolics—on the three wines under investigation allowed us to obtain further insights into their polyphenolic profiles (Table 1). Magliocco showed a total amount of anthocyanins more than twice as abundant as that of either Gaglioppo or Nerello. The amount of total phenolics confirmed that Nerello is the poorest in these important compounds. For the other two autochthonous wines, the amount of low molecular weight phenolics (LMWP) is almost twice as abundant as that of the high molecular weight phenolics (HMWP) (Table 1).

### 2.2. Antioxidant Activity and Phenolic Content

The antioxidant capacity and the reducing power of red wines determined by the TAC (Figure 1a,b) and FPAP assays (Figure 1c,d) showed higher values for Magliocco (20,240 ± 2139 and 22,945 ± 1596, respectively) compared to Nerello (10,123 ± 427 and 10,740 ± 197, respectively) (*p* < 0.01) and Gaglioppo (17,014 ± 590 and 18,447 ± 1632, respectively) (*p* < 0.05). No differences in ferrous equivalents (nmol/L) were observed between the Magliocco and Gaglioppo wines. Of note, significant correlations were observed between the antioxidant activity and biomolecules mainly responsible for the antioxidant properties (Figure 1e) (*p* ≤ 0.05).

### 2.3. Cytoprotective Effects

In vitro assays on the endothelial cell models of insulin resistance and hyperglycemia were used to test the cytoprotective effects of the three red wines studied. To this end, Magliocco (Ma), Gaglioppo (Ga), and Nerello (Ne) were firstly tested for their effects on the cell viability by incubating EC with concentrations of wine up to 10 µg/mL for 24 h, 48 h, and 72 h. The results showed that the cell viability was not affected up to 72 h of treatment (Appendix A Appendix A). When tested in combination with hGlu or PA, coincubation counteracted the negative effects of hGlu and PA on cell viability (*p* < 0.01 vs. control cells) (Appendix A Appendix A). In particular, Ma+hGlu and Ma+PA (Appendix A) showed the highest potency at the concentration of 6 µg/mL after 48 h (*p* < 0.01 vs. hGlu; *p* < 0.01 vs. PA), compared to Ga+hGlu and Ga+PA (Appendix A Appendix A) or Ne+hGlu and Ne+PA (Appendix A). Among the wines tested, Magliocco was found to be more effective in protecting endothelial cells against the cytotoxicity induced by hGlu and PA (Figure 2a,d) (*p* < 0.01 vs. hGlu; *p* < 0.01 vs. PA), compared to Gaglioppo (Figure 2b,e) and Nerello (Figure 2c,f) (*p* < 0.05 vs. hGlu; *p* < 0.05 vs. PA). Based on these results, Magliocco (6 µg/mL), displaying the highest antioxidant and cytoprotective properties, was chosen to investigate the molecular mechanism of endothelial protection.

### 2.4. Anti-Inflammatory and Scavenger Action

The cytoprotective effect of Magliocco was accompanied by its efficacy in reducing IL-6, IL-8, and MCP-1 release induced by hGlu or PA (*p* < 0.001 vs. c Ctr) (Figure 3a–c). Coincubation with Magliocco also reduced TNF-α levels and protein expression (*p* < 0.05 vs. hGlu and *p* < 0.05 vs. PA) (Figure 3d–f). Moreover, the increased production of ROS by PA and hGlu (2.6 and 1.9 times, respectively; *p* < 0.01 vs. Ctr) was inhibited by cotreatment with Magliocco (*p* < 0.01 vs. hGlu and PA) (Figure 3g). Extracellular ROS measurements with the Amplex Red assay confirmed that the exposure to Magliocco was effective in protecting cells from ROS accumulation (*p* < 0.01 vs. hGlu and PA (Figure 3h). Finally, the mitochondrial redox status evaluated by chemical probe MitoSox (Figure 3i,j) showed that the mitochondrial ROS activated by hGlu and PA (*p* < 0.001 vs. Ctr), was consistently reduced by cotreatment with Magliocco (39.3 ± 7.6 vs. 67.89 ± 2.82 AFU in hGlu; 43.68 ± 3.4 vs. 76.09 ± 10.2 AFU in PA).

### 2.5. SIRT1 and SIRT6 Activation

The capacity of Magliocco to modulate SIRT1 and SIRT6 protein expression was further investigated by confocal laser scanner microscopy and Western blot analyses, thus providing cellular localization and protein expression on whole extracts, respectively. The results indicated that Magliocco was opposed to the nuclear downregulation of SIRT1 and SIRT6 induced by hGlu and PA (Figure 4) (*p* < 0.01 vs. Ctr). Specifically, compared with hGlu- and PA-treated cells, SIRT1 arbitrary fluorescence units (AFU) were higher in cells cotreated Ma+hGlu (38.08 ± 4.33 vs. 26.03 ± 2.49 AFU in hGlu, *p* < 0.01) and Ma+PA (35.11 ± 3.55 vs. 26.03 ± 2.49 AFU in PA, *p* < 0.01), as confirmed by Western blot analysis (Figure 4a–e). Moreover, as for SIRT1, an increased SIRT6 expression was observed in cells treated with Ma+hGlu (51.61 ± 3.13 vs. 39.97 ± 4.95 AFU in hGlu) and Ma+PA (41.61 ± 3.77 vs. 29.01 ± 3.855 AFU in PA) (Figure 4f–j). Less consistent effects in counteracting SIRT1 and SIRT6 downregulation occurring under hGlu and PA stressful conditions were observed following Gaglioppo and Nerello treatments (*p* < 0.05 vs. hGlu and *p* < 0.05 vs. PA) (Appendix A Appendix A).

The activation of SIRT1 and SIRT6 was also reflected in the modulation of their downstream targets. Indeed, the increased levels of total and acetylated NF-κB protein expression during treatments with hGlu and PA (*p* < 0.01 vs. Ctr) were reduced by cotreatment with Magliocco (0.6 µg/mL) (0.72 ± 0.062 vs. 1.01 ± 0.115 AU in hGlu; 0.88 ± 0.045 vs. 1.13 ± 0.115 in PA-treated cells) (Figure 5a–g).

Finally, treatments with hGlu and PA caused the increase of total and acetylated p53 levels (*p* < 0.01 vs. Ctr), which were restored by Magliocco coincubation (*p* < 0.05 vs. hGlu and PA alone) (Figure 6a–g).

## 3. Discussion

In the present study, we provide novel evidence on the in vitro bioactivity of red wines in endothelial cell models of insulin resistance and hyperglycemia related to the quantitative and qualitative phenolic profiles. The Italian red wines, Magliocco, Gaglioppo, and Nerello Mascalese, displayed different polyphenol profiles, with Magliocco being the richest in polyphenols and exhibiting the most potent of cytoprotective effect as activator of SIRT1 and SIRT6.

The chemical analysis of red wines was conducted by three different techniques to achieve information on both the major monomeric and polymeric polyphenol contents. Firstly, the untargeted NMR-based analysis was employed to provide a comprehensive picture of the low molecular weight polyphenols, including phenolic acids, flavonoids, and hydroxycinnamic acids. The results revealed that Magliocco displayed the highest concentration of flavan-3-ols and, in particular, of catechin but presented the lowest content of gallic acid. Pyrogallol was quite abundant in both Gaglioppo and Magliocco, while Nerello Mascalese showed the highest content of ethyl caffeate. Structurally, pyrogallol is related to gallic acid, from which it is industrially produced by thermal decarboxylation under drastic conditions of pressure and temperature [15]. The occurrence of pyrogallol in wines has been suggested to be the result of the aging in certain types of wood [16] and is of great interest, as it has been shown to exert lung cancer cell growth inhibition [17], antioxidant, antiseptic, and antipsoriatic properties [18]. Additionally, tyrosol and 2-phenylethanol, produced during the fermentation of tyrosine and 2-phenylalanine, respectively, by the Ehrlich pathway mediated by yeasts, display several health enhancing activities, such as antioxidant, anticancer, antimicrobial, and even cardio-preventive actions [19]. Regarding flavan-3-ols, the detected amounts of catechin and epicatechin evaluated in the analyzed wines are quite common and have important enological impacts. Indeed, flavan-3-ols influence both the organoleptic qualities, as well as biological properties, of wines. As bitter and astringent compounds, flavan-3-ols have been extensively studied to investigate their role in determining the gustative equilibrium of wines. Additionally, catechin and epicatechin are of great pharmacological interest, owing to their many health-related effects, such as the antioxidative, antihypertensive, anti-inflammatory, antiproliferative, antithrombogenic, and antihyperlipidemic ones [20]. The results obtained from the HPLC/UV-Vis analysis allowed the identification and quantification of the anthocyanin contents. These latter, which constitute a class of natural polyphenols responsible for both the quality and longevity of red wines, are native grape pigments responsible for the color of red wine, and their relative amounts are a specific feature of each grape variety and, consequently, of the derived wines [21]. We found that malvidin was the most abundant anthocyanin in Magliocco wine, while a lower percentage of this anthocyanin was detected in Gaglioppo and Nerello. Similar, anthocyanin profiles have been reported for Nerello, while studies conducted on Gaglioppo wine have described a clear dominance of peonidin [14]. We hypothesized that the discrepancy between our data and what was reported in the literature for Gaglioppo could be related to the fact that the Gaglioppo wine we analyzed was, in fact, a blend of Gaglioppo and Magliocco (1:1 *v*/*v*).

The oligomeric and polymeric wine polyphenols analyzed by spectrophotometric investigation also revealed that Magliocco was the richest one in terms of the anthocyanin amount with respect to the other two wines. The formation of polymeric pigments in red wines is a long and slow process modulated by the oxygen uptake, as well as by the initial phenolic composition [22]. Our data suggested that Magliocco abundant anthocyanins likely continue to react, thus further stabilizing the wine color. Indeed, a previous study demonstrated that red wine color stabilization depends on reactive monomeric anthocyanins and on their ratio to LMWP [23,24]. As a consequence, Nerello is poised to improve its color too, but to a lesser extent than Magliocco, while Gaglioppo appears not to be able to further stabilize its pigments because of its quite low concentration of anthocyanins. The content of LMWT determined in Gaglioppo is similar to that reported by Bosso et al. [25], and when LMWT are more abundant than HMWT, wines usually taste less astringent and more bitter. However, the astringency intensity in wine depends on a great number of factors [26] and is strictly correlated to the binding affinity of tannins, whose chemical structures are crucial for their interaction with saliva. Under an enological point of view, the evidence that there is still a large quantity of reactive compounds present in Magliocco, such as monomeric anthocyanins, catechins, and LMWP, suggests that, in this wine, the phenolic profile is likely to evolve over time, thus changing the wine stability, sensory properties, and health-related effects.

The definition of the chemical compositions of Magliocco, Gaglioppo, and Nerello Mascalese paved the way to investigate the possible correlations with their bioactive properties. The antioxidant capacity and reducing power showed a higher biological activity in Magliocco wine, in line with its richest polyphenolic profile (flavonoids, flavonols, total anthocyanins, and polymeric pigments) compared to Nerello and Gaglioppo. The correlation between the phenolic contents of the three analyzed wines with their antioxidant properties underlined that antioxidant activities increased as a function of the degree of polymerization of polyphenols, consistently with what reported by Li and Sun [10]. The Magliocco wine was able to counteract hyperglycemia and insulin resistance, the latter a typical feature of prediabetes status, affecting more than 400 million in the world. Projections indicate that, by 2030, more than 470 million people will suffer from prediabetes [27]. Polyphenols, indeed, have been widely studied for their capacity to reduce insulin resistance by the activation of AMPK (AMP-activated protein kinase) or inhibition of the mTORC1 and PI3K/AkT pathways in several experimental models [28]. Additionally, AMPK activation by polyphenols increases the glucose uptake by positively affecting the endothelial nitric oxide synthase (eNOS) expression and lowering the insulin resistance by inhibiting PI3K/AkT and JNK in the activation of the AMPK-SIRT1-PGC1α axis [28]. The anti-inflammatory effects of Magliocco are probably ascribable to its quite high content of epicatechin, catechin, and flavanols that beneficially impact the endothelial function and prevent cardiovascular diseases by reducing the proinflammatory milieu in TNF-α-activated endothelial cells [29]. Moreover, flavanols and flavonols exert their vascular-protective role by reducing the manifestations of age-related vascular injury. Indeed, they reduce nicotinamide adenine dinucleotide phosphate (NADPH) oxidase by affecting MAPK signaling and downregulating the expression of genes via the p38-MAPK and p65-NF-kB pathways [29]. It is also well-known that excessive ROS production results in the accumulation of DNA damage and induces cellular senescence and that polyphenols can inhibit p53-induced endothelial senescence mediated by TNF-α [30]. In line with these observations, our data showed that Magliocco wine counteracted the increased NF-kB protein expression levels that occurred under high-glucose and palmitic acid stimulation in endothelial cells. Additionally, Magliocco wine suppressed ROS production and inhibited p53 and TNF-α expression in endothelial cells under metabolic stress. It is generally accepted that hyperglycemic vascular complications are associated with oxidative stress and that the endothelial loss of SIRT1 and SIRT6 during oxidative damage is related to an increased expression of total and acetylated NF-κB and p53 [6]. The results from this study also revealed that Magliocco wine prevented oxidative stress via the direct/indirect modulation of SIRT1 and SIRT6, key regulators of the metabolism, DNA repair, and inflammatory response, mainly localized to the nucleus but resulting in cytosolic translocation under stressful conditions.

Polyphenol bioactivities depend on their oral bioavailability, which is usually quite low [31]. Nonetheless, it seems to be high enough to cause polyphenols to exert their health-related beneficial effects altogether. However, as mentioned above, the most abundant grape and wine polyphenols occur as polymeric compounds. In red wines, polymeric polyphenols range from 1000 to 5000 mg/L, as opposed to monomeric compounds such as resveratrol, a widely studied stilbene that is usually present in concentrations hardly above 5 mg/L [32]. The role of resveratrol contained in red wine as an activator in the SIRT1 prevention of endothelial injuries related to atherogenesis and hyperglycemia stress is, to date, widely described [33,34,35]. However, when evaluating the health-related benefits of wines, the contribution of polymeric polyphenols cannot be overlooked. Some studies have indeed ascertained that the radical scavenging activity of proanthocyanidins, measured by the DPPH assay, is positively correlated to their degree of polymerization, with polymeric compounds more active than oligomeric ones that, in turn, are more potent scavengers than monomeric molecules, such as catechins [10]. The protective effectiveness of Magliocco wine is, presumably, also ascribable to the presence of large quantities of reactive compounds, such as monomeric anthocyanins, catechins, and LMWP in this wine. Furthermore, the contribution of HMWP, of which Magliocco is rich, cannot be ruled out, thus finally showing that wine benefits may be accomplished by the simultaneous mixture of compounds rather than from the action of a single class of polyphenols.

## 4. Materials and Methods

### 4.1. Experimental Wines

Monovarietal wines were obtained from Nerello Mascalese and Magliocco grape produced in the Calabria region (Italy) by Marrelli Wines (Le Verdi Praterie Società Agricola a.r.l, Crotone, Italy) in a 2019 vintage, while the blend Gaglioppo-Magliocco (50–50%) was produced in the same area but in a 2018 vintage, as previously reported [11]. As for the standard winemaking process, grapes were destemmed and crushed. They were treated with K_2_S_2_O_5_ (60 mg/Kg of grapes) and 50 mg/kg of ascorbic acid. Fermentation took place at 18–21 °C with indigenous yeast, and the cap was immersed two to four times per day. Maceration of the pomace lasted twelve days. Wines were aged in contact with oak wood. The base parameters of the wines were determined by using official methods of analysis at bottling. For the chemical analysis, two bottles for each treatment were analyzed. Alcohol, residual sugar, titratable acidity, volatile acidity, and free and total sulfur dioxide were determined by the official method of analysis (OIV-MA-F1-07, RESOLUTION OIV-OENO 419A/2011, www.oiv.int, accessed on 16 February 2021).

### 4.2. Analysis of Wine Monomeric Anthocyanins

The analysis of anthocyanins was carried out by high-performance liquid chromatography analysis HPLC-DAD (RESOLUTION OIV-MA-AS315-11, www.oiv.int, accessed on 16 February 2021) [11]. A HPLC SHIMADZU LC10 ADVP apparatus (Shimadzu Italy, Milan), consisting of a SCL-10AVP system controller, two LC-10ADVP pumps, an SPD-M 10 AVP detector, and an injection system full rheodyne model 7725 (Rheodyne, Cotati, CA) equipped with a 50-µL loop, was employed. A Waters Spherisorb column (250 × 4.6 mm, 4 μm particles diameter) with a pre-column was used. Twenty microliters of wine or calibration standard (malvidin-3-O-glucoside chloride, purity >95%, purchased from Sigma-Aldrich, St. Louis, MO, USA) were injected onto the column. Detection was performed by monitoring the absorbance signals at 518 nm. All the samples were filtered through 0.45-micron, durapore membrane filters (Sigma Aldrich, Milan, Italy) into glass vials and immediately injected into the HPLC system. The HPLC solvents were: solvent A: water/formic acid/acetonitrile (87:10:3) *v*/*v*; solvent B: water/formic acid/acetonitrile (40:10:50) *v*/*v*. The gradient used was: zero-time conditions 94% A and 6% B, after 15 min the pumps were adjusted to 70% A and 30% A, at 30 min to 50% A and 50% B, at 35 min to 40% A, and 60% B, at 41 min through the end of the analysis, to 94% A and 6% B. After a 10-min equilibrium period the next sample was injected. The flow rate was 0.80 mL/min. For calibration purposes, the external standard method was used: the calibration curve was plotted for the malvidin-3-O-monoglucoside (Extrasynthese, Lyon, France) on the basis of peak area, and the concentration was expressed as mg/L of malvidin-3-O-glucoside. All the analyses were conducted in duplicate on each experimental replicate.

### 4.3. Analysis of Wine Phenolic Compounds by Spectrophotometry

Total anthocyanins, short polymeric pigments (SPP), large polymeric pigments (LPP), and BSA reactive tannins indicating the high molecular weight phenolics HMWP were determined by the Harbertson-Adams assay, as previously reported [11]. Color intensity, hue, and vanillin reactive flavans, indicating low molecular weight phenolics (LMWP), were determined as described by Gambuti et al. [11,23]. Additionally, total phenolics were also determined by the Harbertson-Adams assay by using a Shimadzu UV-1800 (Kyoto, Japan) UV spectrophotometer.

### 4.4. NMR Analyses of Wines

Aliquots of each wine (100 mL) were concentrated under vacuum in order to remove ethanol and then partitioned twice with ethyl acetate (EtOAc). Twenty milliliters of each EtOAc extract were dried under vacuum, fully solubilized in 0.6 mL of CD_3_OD (99.8% purity; ARMAR Isotopes GmbH, Leipzig, Germany) containing 2.0-mM deuterated 3-(Trimethylsilyl) propionic-2,2,3,3-d4 acid sodium salt (d4-TMSP; min 99 atom%D ARMAR Isotopes GmbH, Leipzig, Germany), and submitted to the NMR-based analysis. Each sample was prepared in duplicate.

NMR spectra were recorded on a Bruker Avance Neo 700 MHz (700 and 175 MHz for 1H and 13C NMR, respectively) using a Norell^®^ Select Series™ 5-mm NMR tubes. Chemical shifts are referenced to the residual solvent signal (CD_3_OD: δ_H_ 3.31, δ_C_ 49.0). Standard Bruker pulse sequences were employed for 1H NMR spectra. d4-TMSP was employed as a calibration standard for NMR-based quantification of analytes. 1H NMR spectra were acquired setting the d1 value at 5.0 s in order to allow a complete relaxation of the d4-TMSP standard to equilibrium [36]. Quantitation of compounds was conducted by selecting representative NMR signals and their areas determined by integration. The molar ratio of each analyte to TMSP was calculated as follows:M”analyte” /M”TMSP” = N”TMSP” /N”analyte” × A”analyte” /A”TMSP”(1)
where M is the molarity, N the number of nuclei generating the investigated NMR signal, and A the peak area. Peak areas (A) were determined by peak picking in combination with line-fitting deconvolution by Mestrenova 9.0 software.

### 4.5. NMR Data of Major Polyphenols Identified in Wines

Catechin. 1H NMR data in CD_3_OD at 25 °C (700 MHz): H2 4.57 (doublet J 7.4 Hz); H3 3.97 (multiplet); H4a 2.50 (doublet of doublets J 16.5, 9.0 Hz); H4b 2.86 (doublet of doublets J 16.5, 1.7 Hz); H6 5.86 (doublet J 2.2 Hz); H8 5.92 (doublet J 2.2 Hz); H2′ 6.84 (doublet J 1.6 Hz); H5′ 6.76 (doublet J 7.1 Hz); H6′ 6.72 (doublet of doublets J 7.1, 1.6 Hz). Catechin was detected in all wines.

Epicatechin. 1H NMR data in CD_3_OD at 25 °C (700 MHz): H3 4.18 (multiplet); H4a 2.74 (doublet of doublets J 16.6, 1.5 Hz); H4b 2.84 (doublet of doublets J 16.6, 2.4 Hz); H6 5.93 (doublet J 2.3 Hz); H8 5.95 (doublet J 2.3 Hz); H2′ 6.97 (doublet J 1.6 Hz); H5′ 6.76 (doublet J 7.1 Hz); H6′ 6.80 (doublet of doublets J 7.1, 1.6 Hz). Epicatechin was detected in all wines.

Ethyl Caffeate. 1H NMR data in CD_3_OD at 25 °C (700 MHz): H2 7.04 (doublet J 1.9 Hz); H5 6.78 (doublet J 8.1 Hz); H6 6.94 (doublet of doublets J 8.1, 1.9 Hz); H7 7.53 (doublet J 15.8 Hz); H8 6.22 (doublet J 15.8 Hz); H21′ 4.20 (quadruplet J 7.1 Hz); H32′ 1.34 (triplet J 7.1 Hz). Ethyl caffeate was detected in Nerello Mascalese and Gaglioppo.

Gallic acid. 1H NMR data in CD_3_OD at 25 °C (700 MHz): H2/H6 7.06 (singlet). Gallic acid was detected in all wines.

2-phenylethanol. 1H NMR data in CD_3_OD at 25 °C (700 MHz): H21 3.74 (triplet J 7.1 Hz); H22 2.81 (triplet J 7.1 Hz); aromatic ring constituted by H2′/6′, H3′/5′, H4′: overlapped resonances centered at 7.17, 7.21 and 7.26. 2-phenylethanol was detected in all wines.

Pyrogallol. 1H NMR data in CD_3_OD at 25 °C (700 MHz): H4/H6 6.32 (doublet J 8.1 Hz); H5 6.49 (triplet J 8.1 Hz). Pyrogallol was detected in all wines.

Tyrosol. 1H NMR data in CD_3_OD at 25 °C (700 MHz): H22/6 7.03 (doublet J 8.4 Hz); H23/5 6.70 (doublet J 8.4 Hz); H27 2.71 (triplet J 7.2 Hz); H28 3.67 (triplet J 7.2 Hz). Tyrosol was detected in all wines.

### 4.6. Antioxidant Assays

The total antioxidant capacity was determined using a colorimetric assay (Abcam, Cambridge, UK, ab65329) based on the Cu^2+^ conversion to Cu^+^ by antioxidants and, to the following release of a colorimetric probe, proportional to the total antioxidant power. The assay was performed following the manufacturer’s instructions. Briefly, samples were mixed with 100 µL of the Cu^2+^ working solution and then incubated for 90 min at room temperature in the dark. The reaction was detected by measuring, with a microplate reader model 680 Bio-Rad (Bio-Rad, Hercules, CA, USA), the absorbance at 570 nm. Recorded absorbances are interpolated with the standard curve of Trolox, a known antioxidant, and the total antioxidant capacity expressed as nM equivalents. The ability of wines to reduce ferric iron (Fe^3+^) to ferrous iron (Fe^2+^) by generating a colorimetric reaction was on the basis of the ferric reducing antioxidant power assay (Abcam, Cambridge, UK, ab234626). The reducing power of the samples was calculated by reacting 10 μL of sample with 190 μL of the reaction mixture and monitoring the increase in absorbance at 594 nm for 1 h at 37 °C. The antioxidant potential of samples was determined using a ferrous iron standard curve and results are expressed as Fe^2+^ equivalents (nM).

### 4.7. Cell Culture and Treatment

Endothelial cells (EC) were purchased from American Type Culture Collection (ATCC, Manassas, VA, USA, CCL 209). Cells were maintained in minimum essential medium (MEM, Gibco, Life Technologies, Carlsbad, CA, USA, 11095-080), supplemented with 20% heat-inactivated fetal bovine serum, penicillin (100 U/mL), and streptomycin (0.1 mg/mL) at 37 °C in a humidified atmosphere, 95% air, 5% CO_2_. Exposure to high-glucose (hGlu) (30 mM) and palmitic acid (PA) (0.5 mM) were used as a pro-oxidant and inflammatory stimuli [37,38,39,40] to mimic oxidative microenvironment. In order to exclude any interference with ethanol, cell treatments were performed using lyophilized wine samples dissolved in Hanks’ balanced salt solution (HBSS)–10 mM of Hepes (0–10 µg/mL). Cells were pretreated with wine samples for 12 h before exposure to hGlu (30 mM) or PA (0.5 mM) added to the same culture medium. The coincubation of wine + hGlu (30 mM) or wine + PA (0.5 mM) was performed for 24 h, 48 h, and 72 h at 37 °C in a humidified atmosphere, 95% air, 5% CO_2_. Control cells (Ctr) were treated with corresponding volumes of Hanks’ balanced salt solution (HBSS)–10 mM of Hepes.

### 4.8. Assessment of Cell Viability and Cytotoxicity

Cell viability was detected using Cell Counting Kit-8 (CCK-8 Donjindo Molecular Technologies, Inc., Rockville, MD, USA) following the manufacturer’s instructions, as previously described [41]. Briefly, 10 μL of CCK-8 solution was added to each well, and then, the cells were incubated at 37 °C for 4 h. Thereafter, the absorbance was measured at 450 nm using a microplate reader model 680 Bio-Rad (Bio-Rad, Hercules, CA, USA). Cellular membrane integrity was assessed using cytotoxicity LDH Assay Kit-WST (Donjindo Molecular Technologies, Inc., Rockville, MD, USA, CK12). The release of lactate dehydrogenase (LDH) into the medium was measured according to the manufacturer’s instructions. Briefly, after treatments, 100 μL of the working solution was added to 50 μL of endothelial cell suspension. The 96-well culture plate was then incubated at room temperature for 30 min protected from light. Sample absorbance was determined at 490 nm on a microplate reader model 680 Bio-Rad (Bio-Rad, Hercules, CA, USA), and the percent of cytotoxicity was calculated by the following equation: (substance—low control)/(high control—low control) × 100. All experiments were performed with *n* = 5 replicates.

### 4.9. Cell Lysates and Western Blotting

After the above-mentioned treatments, the cellular total protein content was extracted, as previously described [42], and the total protein concentration was determined by the Bio-Rad Protein Assay kit (Bio-Rad, Hercules, CA, USA). According to the target protein band size, protein samples were subjected to sodium dodecyl sulfate-polyacrylamide gel electrophoresis (SDS-PAGE) (8–12%) and transferred to nitrocellulose membranes (Bio-Rad, Hercules, CA, USA). Membranes were blocked in 10-mM Tris-HCl, pH 8.0, 150-mM NaCl, and 0.05% Tween 20 (TBST) supplemented with 5% nonfat dry milk for 1 h at room temperature. Membranes were incubated overnight at 4 °C with specific primary antibodies anti-SIRT1 (1:1000, Biorbyt, Cambridge, UK, orb306144), anti-SIRT6 (1:1000, Abcam, Cambridge, UK, ab191385), anti-NF-κB p65 (acetyl K310) (1:1000, Abcam, Cambridge, UK, ab218533), anti-NF-κB (1:1000, Abcam, Cambridge, UK, ab16502), anti-p53 (acetyl K382) (1:700, Abcam, Cambridge, UK, ab75754), anti-p53 (1:1000, Biorbyt, Cambridge, UK, orb323871), and anti-tumor necrosis factor-α (TNF-α) (1:1000, Abcam, Cambridge, UK, ab6671). Anti-α-tubulin (1:5000, Cell Signaling Technology, Danvers, MA, USA, 3873), anti-β-actin (1:5000, Cell Signaling Technology, Danvers, MA, USA, 3700), and anti -GAPDH (1:10.000, Abcam, Cambridge, UK, ab9485) were used as loading controls. After 1 h incubation with HRP-conjugated secondary antibodies (NC GxMu-003-DHRPX and GtxRb-003-DHRPX, ImmunoReagents Inc., Raleigh, NC, USA), the immunocomplexes were examined by the Excellent chemiluminescent sustrate kit (Elabscience Biotechnology Inc., Houston, TX, USA, E-IR-R301) and visualized by using the ChemiDoc Imaging System with Image Lab 6.0.1 software (Bio-Rad Laboratories, Milan, Italy). The analyses of immunoblotting data were performed with ImageJ 1.52n software (National Institutes of Health) by measuring the density of each band and, after background subtraction, comparing it with the loading control signal. Results were reported as arbitrary units (AU) ± SD of at least three independent experiments.

### 4.10. Intracellular ROS Detection

Endothelial cells (5 × 10^3^ cells/well) were seeded in a 96-well microplate. Intracellular ROS levels were determined by using the cellular reactive orange fluorescence oxygen species detection assay kit (Abcam, Cambridge, UK, ab186028), as previously described [43]. Briefly, 100 μL of ROS working solution was added to each well, and the colorimetric reaction proceeded for 60 min until the fluorescence intensity was measured at an excitation wavelength of 540 nm and an emission wavelength of 570 nm using a Tecan Infinite 2000 Multiplate reader (Tecan, Männedorf, Swiss).

### 4.11. Extracellular ROS Evaluation

To evaluate the extracellular H_2_O_2_ released from endothelial cells, the Amplex Red Hydrogen Peroxide/Peroxidase Assay Kit (Thermo Fisher Scientific, Waltham, MA, USA, A22188) was used as previously described [42]. Briefly, the suspension of live cells (2 × 10^4^) prepared in a Krebs–Ringer phosphate glucose buffer (145 mM NaCl, 5.7 mM sodium phosphate, 4.86 mM KCl, 0.54 mM CaCl_2_, 1.22 mM MgSO_4_, and 5.5 mM glucose, pH 7.35) was mixed with 100 μL Amplex Red reagent containing 50 μM Amplex Red and 0.1-U HRP/mL. After a 60 min incubation at 37 °C, the fluorescence was measured at excitation wavelength of 530 nm and emission wavelength of 590 nm, using a Tecan Infinite 2000 Multiplate reader (Tecan, Männedorf, Swiss). Extracellular ROS content was calculated interpolating the sample fluorescence values with an H_2_O_2_ standard curve (0–2 μM concentration range).

### 4.12. Mitochondrial ROS Measurement

To detect the generation of mitochondrial ROS, the Mitosox Red Mitochondrial Superoxide Indicator (Thermo Scientific, Rockford, IL, USA, M36008) was used as indicated by its manufacturer’s protocol. EC were seeded in 24-well plate containing microscope glass (12 mm) (Thermo Fisher Scientific, Waltham, MA, USA) and stained for 10 min with 5-µM Mitosox at 37 °C before the paraformaldehyde fixing, as reported in the “Confocal laser scanning microscopy” section. The superoxide levels were assessed by confocal laser microscopy, and the fluorescence intensity, expressed as Arbitrary Fluorescence Units (AFU), was calculated with ImageJ1.52n software (National Institutes of Health, Bethesda, MD, USA). Menadione (50 μM) (Sigma Aldrich, St. Louis, MO, USA, M57405) was the positive control.

### 4.13. Assessment of Cytokine Levels

Cytokines (IL-6, IL-8, and TNF-α) and monocyte chemoattractant protein 1 (MCP-1) levels were determined by ELISA assays (human interleukin-6 ELISA, BioVendor Laboratorni medicina a.s., Brno, Czech Republic, RD194015200R; human interleukin-8, BioVendor, Laboratorni medicina a.s., Brno, Czech Republic, RD194558200R; ELISA Cymax TNF-alpha ELISA, AbFrontier, Seoul, Korea, YIF-LF-EK0193; human MCP-1 ELISA, BioVendor, Laboratorni medicina a.s., Brno, Czech Republic, RAF081R, respectively), according to the manufacturer’s instructions. Briefly, 100 μL of EC lysates were incubated in microplate wells precoated with specific anti-cytokine antibodies. After 60 min incubation and washing to remove non-bound cytokines and other components of the sample, biotin-labeled anti-IL-6, -IL-8, -TNF-α, and -MCP-1 antibodies were added and incubated for additional 60 min. After another washing, streptavidin-HRP conjugate is added, following 30-min incubation. The remaining conjugate is allowed to react with the substrate solution, and then, absorbance was measured at 450 nm using a microplate reader model 680 Bio-Rad (Bio-Rad, Hercules, CA, USA). Concentrations of cytokines in samples were resulted by plotting absorbance values against concentrations of each standard curve.

### 4.14. Confocal Laser Scanning Microscopy

Confocal laser scanning microscopy analysis was performed as previously reported [41]. After treatments and Mitosox Red staining, EC were fixed with 4% (*v*/*v*) paraformaldehyde solution for 20 min and then permeabilized with 0.1% (*v*/*v*) Triton X-100 in PBS for 10 min at room temperature. For SIRT1 (1:500, Abcam, Cambridge, UK, ab32441), SIRT6 (1:500, Abcam, Cambridge, UK, ab191385), p53 (1:500, Biorbyt, Cambridge, UK, orb304644), and NF-κB (1:500, Cell Signaling Technology, Danvers, MA, USA, C22B4) immunofluorescence detection, primary antibodies were incubated overnight at 4 °C, followed by incubation with Alexa Fluor 633 (1:1000, Life Technologies, Carlsbad, CA, USA) for 1 h. For immunofluorescence study, anti-vimentin antibody (1:1000, Sigma Aldrich, St. Louis, MO, USA, V6630) was used to stain cytoskeleton, followed by Alexa Fluor 488 (1:1000, Life Technologies, Carlsbad, CA, USA) secondary antibody incubation. As for Mitosox Red staining, cellular architecture was marked by Phalloidin 488 (1:1000, Abcam, Cambridge, UK, ab176753). The nuclear staining, performed for 7 min with 2.5-µg/mL 4′, 6-diamidino-2-phenylindole (DAPI; Sigma Aldrich, St. Louis, MO, USA), was included to samples involved in mitochondrial stress investigation. Microscopy analyses were performed with a LSM 700 confocal microscope (Zeiss, Oberkochen, Germany) equipped with a plan apochromat X63 (NA1.4) oil immersion objective, and fluorescence intensity values were evaluated with ImageJ 1.52n software (National Institutes of Health, Bethesda, MD, USA).

### 4.15. Statistical Analysis

For the biological assays and in vitro experimentation, all reported data referred to at least three independent experiments. Values are presented as the mean ± standard deviation (SD). Results were statistically evaluated using one-way ANOVA, followed by Bonferroni’s post-hoc test. *p* < 0.05 was assumed as a statistically significant difference. For correlations between antioxidant properties and phenolic contents, all data were processed in the R environment. Therefore, Spearman’s correlation coefficients were computed with the rcorr function to measure the strength and direction of association between the variables. The correlation plot was generated by the corrplot package [44].

## 5. Conclusions

The results of this study supported the relevance of both the content and type of red wine phenols in relation to their in vitro efficacy against endothelial oxidative damages induced by insulin resistance and hyperglycemia. Besides the biological relevance of monomeric anthocyanins, catechins, and LMWP, additional mechanistic in vitro and in vivo studies are necessary to fully define their specific roles and the occurrence of additive or synergistic effects as SIRT1 and SIRT6 activators in the prevention of endothelial damage due to type 2 diabetes.

## Figures and Tables

**Figure 1 ijms-22-05677-f001:**
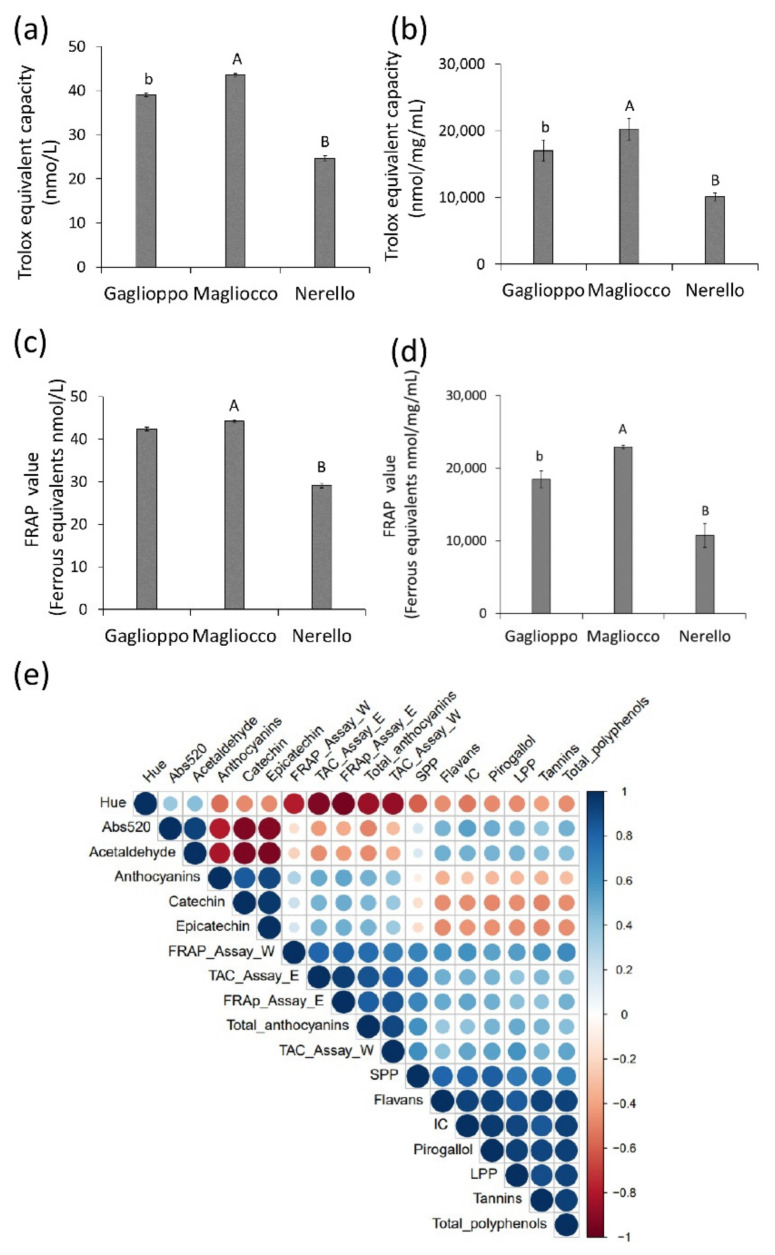
Antioxidant potential of red wines and correlation with the polyphenol contents. Total antioxidant activity and ferric reducing power were reported for (**a,c**) whole red wine samples and (**b,d**) lyophilized wine samples of Gaglioppo, Magliocco, and Nerello. Values represent the mean ± SD of *n* = 3 independent experiments. (**e**) Correlation plot is between the antioxidant activity and the phenolic contents of the wines. Stronger correlations are depicted by larger circles, with shades of blue representing positive correlations and shades of red representing negative correlations. Ab, *p* < 0.05 and AB, *p* < 0.01.

**Figure 2 ijms-22-05677-f002:**
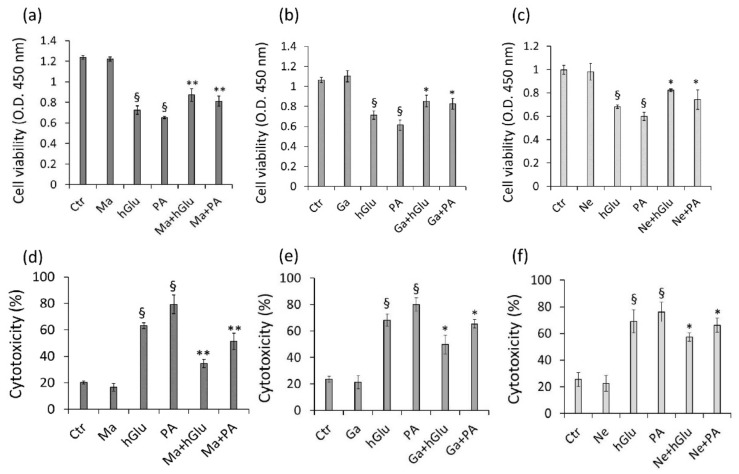
Endothelial protective effects of red wines during insulin resistance and hyperglycemia. (**a**–**c**) Cell viability and (**d**–**f**) cytotoxicity were evaluated by pretreating (12 h) EC with Ma, Ga, or Ne (6 µg/mL) before exposure to hGlu (30 mM) or PA (0.5 mM) for 48 h. Control cells were treated with corresponding volumes of Hanks’ balanced salt solution (HBSS)–10 mM of Hepes. Cell viability was assessed by Cell Counting Kit-8 (Donjindo Molecular Technologies, Inc., Rockville, MD, USA). Cytotoxicity by LDH Assay Kit-WST (Donjindo Molecular Technologies, Inc., Rockville, MD, USA) and expressed as the mean ± SD of *n* = 5 replicates. ^§^
*p* < 0.01 vs. Ctr, * *p* < 0.05 vs. hGlu or PA, and ** *p* < 0.01 vs. hGlu or PA.

**Figure 3 ijms-22-05677-f003:**
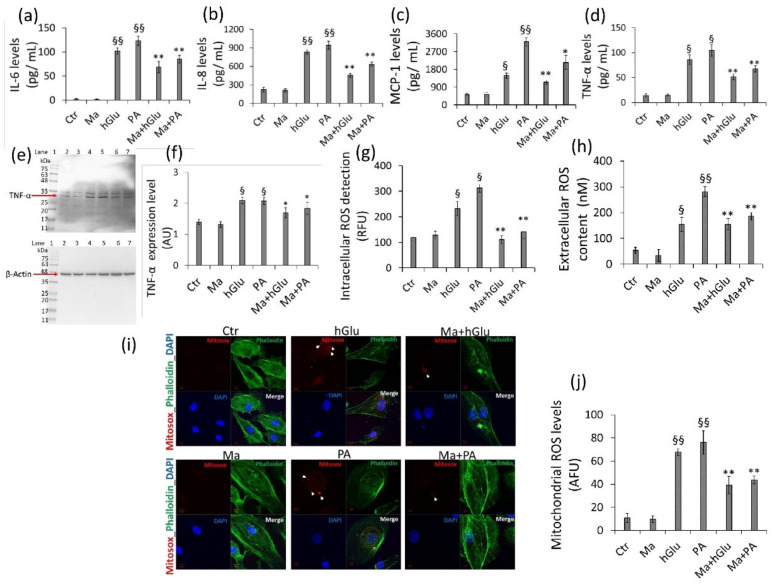
Antioxidant and anti-inflammatory activities of Magliocco. Endothelial cells were exposed to Magliocco (Ma), high glucose (hGlu) (30 mM), palmitic acid (PA) (0.5 mM), or combined Ma+hGlu and Ma+PA for 48 h. (**a**–**d**) IL-6, IL-8, MCP-1, and TNF-α-level measurements. (**e**–**f**) TNF-α protein expression. Lane 1 = protein ladder molecular weight markers, lane 2 = Ctr, lane 3 = Ma, lane 4 = hGlu, lane 5 = PA, lane 6 = Ma+hGlu, and lane 7 = Ma+PA. The analysis of densitometric intensity was calculated with ImageJ software and expressed as arbitrary units (AU) ± SD of *n* = 4 replicates. β-Actin was used as internal control. (**g**) Intracellular and (**h**) extracellular ROS content evaluations. (**i**) Representative images of confocal laser scanning analyses of mitochondrial ROS generation (indicated with white arrows) detected by MitoSOX probe and (**j**) mitochondrial superoxide level assessment. The results are expressed as arbitrary fluorescence units (AFU) ± SD of *n* = 3 replicates. Scale bars = 10 µm. The cytoskeleton is marked with Phalloidin 488 (green), while DAPI was used as a nuclei counterstain (blue). ^§^
*p* < 0.01 vs. Ctr, ^§§^
*p* < 0.001 vs. Ctr, * *p* < 0.05 vs. hGlu or PA, and ** *p* < 0.01 vs. hGlu or PA.

**Figure 4 ijms-22-05677-f004:**
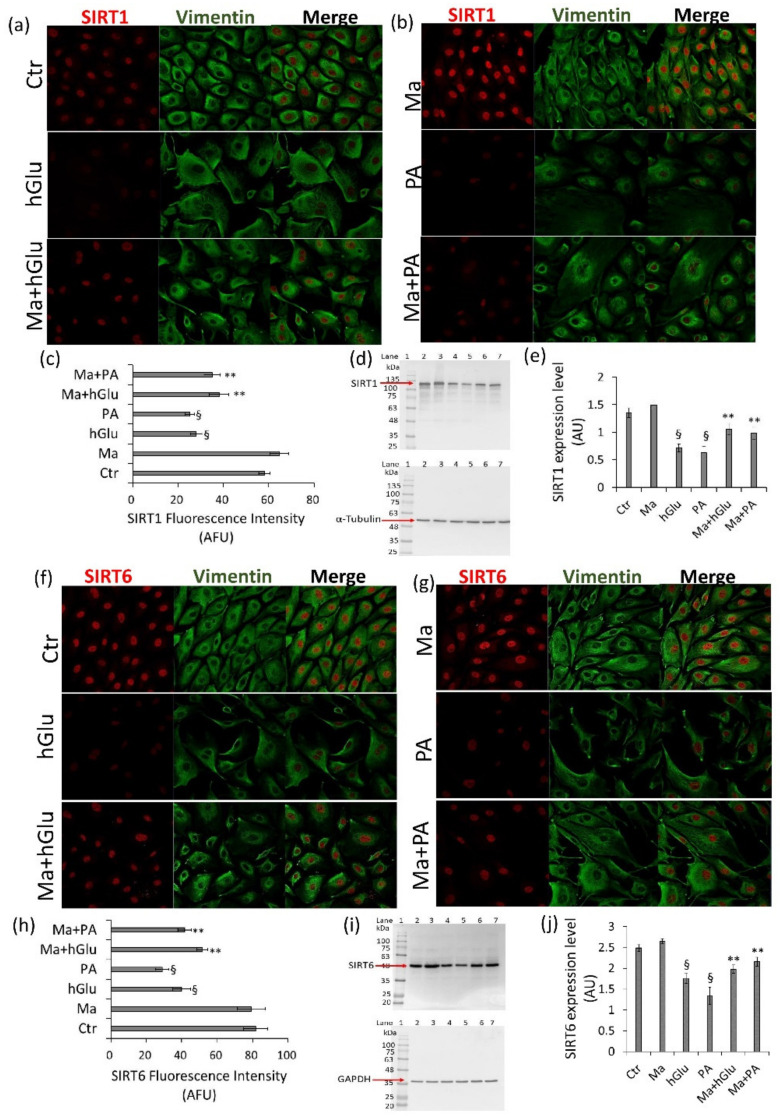
SIRT1 and SIRT6 activation by Magliocco. (**a**,**b**) Representative confocal images of SIRT1 (red) and vimentin (green) and (**c**) fluorescence intensity determination performed by ImageJ software and expressed as arbitrary fluorescence units (AFU) ± SD of *n* = 3 replicates. (**d**,**e**) Western blot analysis of SIRT1 expression levels. (**f**,**g**) Representative confocal images of SIRT6 (red) and vimentin (green). (**h**) Fluorescence intensity analysis. (**i**,**j**) Western blot analysis of the SIRT6 expression levels. Lane 1 = protein ladder molecular weight markers, lane 2 = Ctr, lane 3 = Ma, lane 4 = hGlu, lane 5 = PA, lane 6 = Ma+hGlu, and lane 7 = Ma+PA. The analysis of densitometric intensity was calculated with ImageJ software and expressed as arbitrary units (AU) ± SD of *n* = 4 replicates. α-Tubulin or GAPDH was used as the internal control. ^§^
*p* < 0.01 vs. Ctr, ** *p* < 0.01 vs. hGlu and vs. PA.

**Figure 5 ijms-22-05677-f005:**
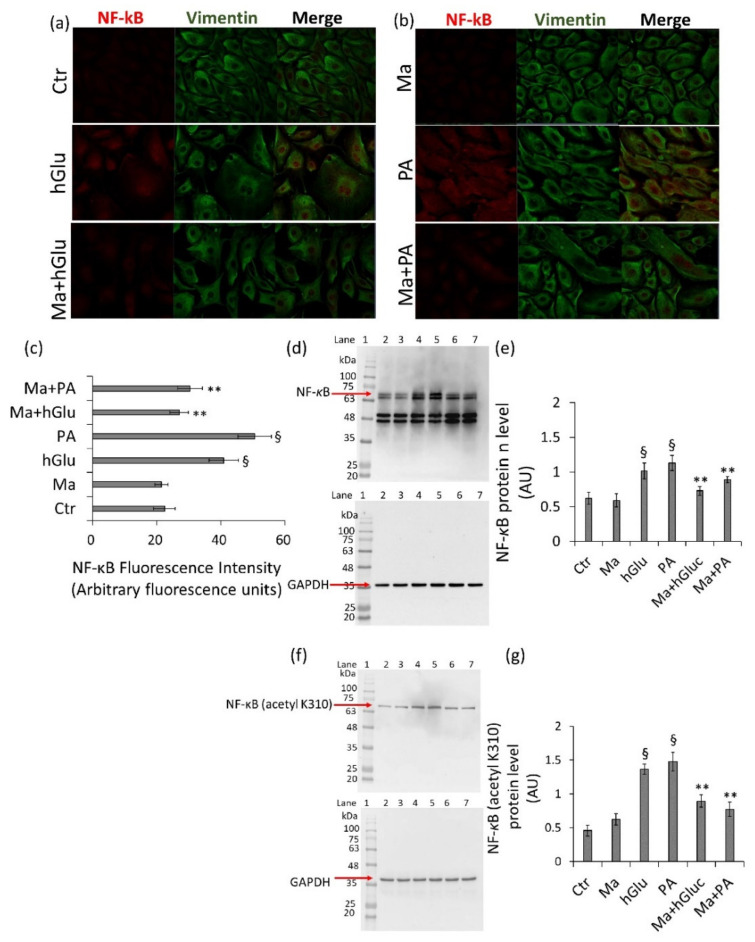
Modulation of NF-κB by Magliocco. (**a**,**b**) Representative confocal images of NF-κB (red) and vimentin (green) and (**c**) fluorescence intensity analysis performed by ImageJ software and expressed as arbitrary fluorescence units (AFU) ± SD of *n* = 4 replicates. (**d**,**e**) Western blot analysis of total and (**f**,**g**) acetylated (acetyl K310) NF-κB expression levels. Lane 1 = protein ladder molecular weight markers, lane 2 = Ctr, lane 3 = Ma, lane 4 = hGlu, lane 5 = PA, lane 6 = Ma+hGlu, and lane 7 = Ma+PA. The analysis of densitometric intensity was calculated with ImageJ software and expressed as arbitrary units (AU) ± SD of *n* = 3 replicates. GAPDH was used as the internal control. ^§^
*p* < 0.01 vs. Ctr, and ** *p* < 0.01 vs. hGlu or PA.

**Figure 6 ijms-22-05677-f006:**
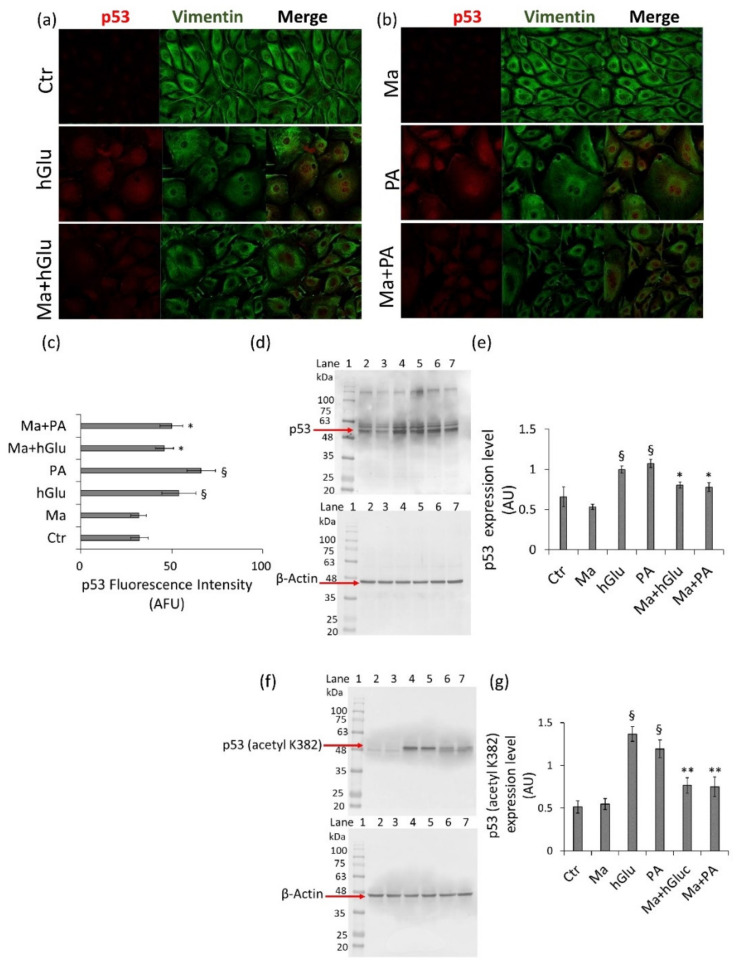
Modulation of p53 by Magliocco. (**a**,**b**) Representative confocal images of p53 (red) and vimentin (green). (**c**) Fluorescence intensity performed by ImageJ software and expressed as arbitrary fluorescence units (AFU) ± SD of *n* = 3 replicates. (**d**,**e**) Western blot analysis of total and (**f**,**g**) acetylated (acetyl K382) p53 expression levels. Lane 1 = protein ladder molecular weight markers, lane 2 = Ctr, lane 3 = Ma, lane 4 = hGlu, lane 5 = PA, lane 6 = Ma+hGlu, and lane 7 = Ma+PA. The analysis of densitometric intensity was calculated with ImageJ software and expressed as arbitrary units (AU) ± SD of *n* = 3 replicates. β-Actin was used as the internal control. ^§^
*p* < 0.01 vs. Ctr, * *p* < 0.05 vs. hGlu or PA, and ** *p* < 0.01 vs. hGlu or PA.

**Table 1 ijms-22-05677-t001:** Chemical characterization of wines. Polyphenols in Gaglioppo–Magliocco, Magliocco, and Nerello were identified by NMR. Concentration of monomeric anthocyanins were determined by HPLC analysis; total phenolics were determined by spectrophotometric methods. Their quantifications are expressed as mg/L. All data were expressed as the mean ± standard deviation (SD) of three replicates. Statistical analysis was performed through an analysis of variance (ANOVA), followed by the Tukey’s multiple comparisons. Statistical significance was attributed to *p*-values < 0.05.

	Nerello Mascalese	Magliocco	Gaglioppo–Magliocco
NMR analysis			
**Polyphenols (mg/L)**			
Pyrogallol	18.6 ± 0.5	96.1 ± 2.2	105.2 ± 2.5
Gallic Acid	62.1 ± 1.3	19.7 ± 0.7	81.8 ± 1.9
Tyrosol	77.2 ± 3.2	65.6 ± 3.7	78.9 ± 2.2
Ethyl Caffeate	81.2 ± 4.1	N. D.	40.2 ± 0.3
Catechin	32.2 ± 1.7	170.0 ± 2.5	74.1 ± 0.8
Epicatechin	10.2 ± 0.2	72.3 ± 2.6	37.0 ± 1.2
HPLC analysis			
**Monomeric anthocyanins (mg/L)**			
Dp3glc	1.73 ± 0.14	9.22 ± 0.94	11.49 ± 0.63
Cy3glc	0.76 ± 0.11	2.49 ± 0.25	5.66 ± 0.61
Pt3glc	2.67 ± 0.56	15.08 ± 3.18	12.22 ± 0.54
Pn3glc	2.78 ± 0.06	11.39 ± 0.62	11.91 ± 1.62
Mv3glc	25.53 ± 0.74	238.39 ± 1.40	52.02 ± 2.13
Mv3acglc	1.02 ± 0.05	9.57 ± 0.32	0.64 ± 0.24
Mv3cmglc	1.76 ± 0.04	24.69 ± 1.29	0.69 ± 0.28
Spectrophotometric analysis			
**Total phenolics (mg/L)**	4722 ± 252	4291 ± 122	3083 ± 122

## Data Availability

The data presented in this study are available from the corresponding author upon request.

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
