# Peer review of "Phenolic Profiles of Red Wine Relate to Vascular Endothelial Benefits Mediated by SIRT1 and SIRT6"

_ijms, 2021, doi:10.3390/ijms22115677_

Round 1
Reviewer 1 Report
D'Onofrio, et al. reported that the phenolic profiles of red wines differ from each other and benefit the cells under oxidative and pro-inflammatory stress with upregulating SIRT1 and SIRT6 expression in endothelial cells. This study provides certain evidence in demonstrating the mechanisms of wine in protecting cells against oxidative stress. However, some portions of the experiment design and results interpretation are not clear, which need to be clarified.
1, Result 2.1, the chemical characterization of three wines using NMR/HPLC-UV/Vis, and the spectrophotometric analysis were stated to use two bottles of each wine on page 14. It is not convincing to just have two replicates for the assay. Meanwhile, there is no statistical analysis for the polyphenols and monomeric anthocyanins in Table 1. However, the content of certain chemicals, such as Pyrogallol, Catechin, and Mv3glc are different among wine types. It is suggested to perform more sample replicates and perform statistical analysis to get the conclusion stated in this study.
2, The authors used the t-test for multi-group experiments, which is inappropriate. There is no clarification in any figures of the replication number for the statistical analysis. Please state the N for each group in all bar graphs.
Besides, there are a few minor issues.
1, There are several grammar issues I have notices, line 17 Here, characterized...; line 25 findings....; line 68 we sought we sought... Please carefully read through the manuscript and correct those errors.
2, It is hard to interpret the results in table 1 with the context stated on page 3. Mv-3-cum and Mv-3-ace were not listed clearly in table 1. It would be better if table 1 is modified with sub-category groups, such as monomeric anthocyanins and Malvidin. Besides, the percentage of each sub-groups should be listed in the table for comparison purposes.
Author Response
Reviewer 1
D'Onofrio, et al. reported that the phenolic profiles of red wines differ from each other and benefit the cells under oxidative and pro-inflammatory stress with upregulating SIRT1 and SIRT6 expression in endothelial cells. This study provides certain evidence in demonstrating the mechanisms of wine in protecting cells against oxidative stress. However, some portions of the experiment design and results interpretation are not clear, which need to be clarified.
Comment: Result 2.1, the chemical characterization of three wines using NMR/HPLC-UV/Vis, and the spectrophotometric analysis were stated to use two bottles of each wine. It is not convincing to just have two replicates for the assay. Meanwhile, there is no statistical analysis for the polyphenols and monomeric anthocyanins in Table 1. However, the content of certain chemicals, such as Pyrogallol, Catechin, and Mv3glc are different among wine types. It is suggested to perform more sample replicates and perform statistical analysis to get the conclusion stated in this study.
Response: According to the Reviewer’s suggestion, statistical analysis has been added to Table 1. See lines 102-104.
Comment: The authors used the t-test for multi-group experiments, which is inappropriate. There is no clarification in any figures of the replication number for the statistical analysis. Please state the N for each group in all bar graphs.
Response: As requested, a detailed explanation of statistical analysis was added in Material and Methods section in paragraph 4.15, “Statistical analysis”, see lines 607-614. The replication number for the statistical analysis was also added in each figure legend.
Besides, there are a few minor issues.
Comment: There are several grammar issues I have notices, line 17 Here, characterized...; line 25 findings....; line 68 we sought we sought... Please carefully read through the manuscript and correct those errors.
Response: Thank you so much for your observations. As requested, we read through the text and tried to amend all the English grammar issues occurred in the revised manuscript.
Comment: It is hard to interpret the results in table 1 with the context stated on page 3. Mv-3-cum and Mv-3-ace were not listed clearly in table 1. It would be better if table 1 is modified with sub-category groups, such as monomeric anthocyanins and Malvidin. Besides, the percentage of each sub-groups should be listed in the table for comparison purposes.
Response: We thank the Reviewer for this observation. As a matter of fact, there was no correspondence between the abbreviations used in the text and those displayed in Table 1. This point has been addressed according to Reviewer comment and Table 1 adjusted accordingly (see new Table 1). We did not provide percentages, as these would not be totally realistic as for each metabolite subgroup. We listed only those molecules that were not falling below the detection thresholds of the employed techniques. This implies that there might be numerous other metabolites, even though minor, in the analysed wines that were not either identified or quantified by us.

Reviewer 2 Report
The authors analyzed the composition of phenolic compounds in three red wines, Gaglioppo, Magliocco and Nerello Mascalese, from various angles using NMR, HPLC and spectrophotometry, and found that Magliocco wine had the highest content of phenolic compounds. We were able to derive that Magliocco wine had the highest content of phenolic compounds. In vitro experiments using the three wines and endothelial cells successfully revealed that Magliocco wine had the highest redox capacity and the highest endothelial cell protection against hyperglycemia and palmitic acid-induced stress without overloading the cells.
The authors further analyzed Magliocco wine and showed that it inhibited the increased production of ROS and mitochondrial abnormalities in endothelial cells caused by metabolic stress. SIRT1/SIRT6 was shown to be the pathway, and Magliocco wine actually improved the expression of SIRT1/SIRT6, which was decreased under stress. In addition, they showed that the downstream expression of NF-kB and p53 protein expression levels were also suppressed.
・The figures for TAC and FRAP in Figure 1 are not given in the text of numerical value, making it difficult to understand the specific values.
・There seems to be a mistake in Figure 2 (b) and (c). Please correct them.
・I don't know much about wine, so I was wondering if you could help me. Why did the authors choose Gaglioppo, Magliocco, and Nerello Mascalese for the study? The authors' previous report (Ref. 11) used five different wines. I would like to know why they chose these three wines.
・We know from the previous figure that Magliocco has the best antioxidant capacity.However, I think it would be better to evaluate the inflammation-related proteins, mitochondrial function, sirtuin expression, and sirtuin target proteins of other Gaglioppo and Nerello Mascalese. If the authors are correct, would any of them be inferior to Magliocco?
・I think that the expression of SIRT1 and SIRT6 as target proteins and NF-kB/p53 expression alone is not enough. If the title of the article is "Mediated by SIRT1 and SIRT6", I would like to see the evaluation of deacetylation ( immunoprecipitation or western blot of the target acetylated protein).
Author Response
Reviewer 2
The authors analyzed the composition of phenolic compounds in three red wines, Gaglioppo, Magliocco and Nerello Mascalese, from various angles using NMR, HPLC and spectrophotometry, and found that Magliocco wine had the highest content of phenolic compounds. We were able to derive that Magliocco wine had the highest content of phenolic compounds. In vitro experiments using the three wines and endothelial cells successfully revealed that Magliocco wine had the highest redox capacity and the highest endothelial cell protection against hyperglycemia and palmitic acid-induced stress without overloading the cells.
The authors further analyzed Magliocco wine and showed that it inhibited the increased production of ROS and mitochondrial abnormalities in endothelial cells caused by metabolic stress. SIRT1/SIRT6 was shown to be the pathway, and Magliocco wine actually improved the expression of SIRT1/SIRT6, which was decreased under stress. In addition, they showed that the downstream expression of NF-kB and p53 protein expression levels were also suppressed.
Comment: The figures for TAC and FRAP in Figure 1 are not given in the text of numerical value, making it difficult to understand the specific values.
Response: As requested, the numerical values of TAC and FRAP assay were added in the Results section, paragraph 2.2 “Antioxidant activity and Phenolic content”. See lines 139-143.
Comment: There seems to be a mistake in Figure 2 (b) and (c). Please correct them.
Response: We thank the Reviewer for reporting the mistake. As requested, the mistake in Figure 2 b and Figure 2 c was adjusted. Please see new Figure 2 b and c.
Comment: I don't know much about wine, so I was wondering if you could help me. Why did the authors choose Gaglioppo, Magliocco, and Nerello Mascalese for the study? The authors' previous report (Ref. 11) used five different wines. I would like to know why they chose these three wines.
Response: As requested, we explained the reason of the choice. New paragraph has been added reporting that “In the present study, we investigated the relationship between the phenolic content of red wine and their in vitro bioactivity. To this end, wines were selected on the basis of the comparable pedoclimatic conditions and the same winemaking protocol. This latter was instrumental in standardizing the wine production procedure in order to rule out any technological variables and allow a straightforward correlation between the bioactivities of the wines and their specific content of healthy natural metabolites. The Italian red wines chosen were Magliocco, Gaglioppo, and Nerello Mascalese, all produced in Calabria region (Italy) and with the same winemaking protocol” (Line 68-75).
The Aglianico and Barbera wines used in the previous study (ref. 11) were not used as they have been produced with a winemaking protocol different from Gaglioppo, Magliocco and Nerello.
Comment: We know from the previous figure that Magliocco has the best antioxidant capacity. However, I think it would be better to evaluate the inflammation-related proteins, mitochondrial function, sirtuin expression, and sirtuin target proteins of other Gaglioppo and Nerello Mascalese. If the authors are correct, would any of them be inferior to Magliocco?
Response: We thank the Reviewer for the observation.
As suggested, we performed new experiments to test the effects of Gaglioppo and Nerello Mascalese on SIRT1 and SIRT6 protein modulation. Results indicate that Gaglioppo and Nerello Mascalese show lower effects compared to Magliocco. See new Supplementary Figure S3. The less consistent effect of Gaglioppo and Nerello treatment on SIRT1 and SIRT6 modulation was also highlighted in the Results section, paragraph 2.5 “SIRT1 and SIRT6 activation”. See lines 218-221.
Comment: I think that the expression of SIRT1 and SIRT6 as target proteins and NF-kB/p53 expression alone is not enough. If the title of the article is "Mediated by SIRT1 and SIRT6", I would like to see the evaluation of deacetylation (immunoprecipitation or western blot of the target acetylated protein).
Response: As suggested, new experiments are performed to evaluate the NF-kB and p53 acetylated protein levels by western blot assay. See new Figure 5f, g and Figure 6f, g.

Reviewer 3 Report
D’Onofrio et al demonstrated an interesting result about antioxidant property of various wines in the manuscript titled “Phenolic profiles of red wine relate to vascular endothelial ben3 efits mediated by SIRT1 and SIRT6” Although, the results are impressive, some raised issues should be resolved.
Major
- The composition of wine was detected by HPLC. Can this result be a representative result of all wines made with this variety? Isn’t there any possibility that it could be a difference between the wines? What methods have you considered to overcome those differences?
- All the representative pictures of western blot analysis should be more clearly demonstrated. The exact methods for the analysis of densitometric intensity should be described in details how arbitrary units (AU) is calculated.
- In the western blot analysis, all the measurements were done using whole cell lysate? The location of each component looks different. For example, NF-KB showed cytosolic expression, whereas SIRT 1 and SIRT 6 were expressed mainly in nuclear location. It should better be done using specific section, which the material were mainly expressed.
- In figure 3, mitosox expression in each figure examined by confocal microscopy was not clearly demonstrated. Using arrows might be helpful for clarification. Moreover, detailed method for measurement of specific material expression by confocal laser microscopy representing AU should be described.
- SIRT1 is known to be shuttled to the cytoplasm under specific stressful conditions. Wasn’t it the case in this experiment?
Author Response
Reviewer 3
Comments and Suggestions for Authors
D’Onofrio et al demonstrated an interesting result about antioxidant property of various wines in the manuscript titled “Phenolic profiles of red wine relate to vascular endothelial benefits mediated by SIRT1 and SIRT6” Although, the results are impressive, some raised issues should be resolved.
Major
- Comment: The composition of wine was detected by HPLC. Can this result be a representative result of all wines made with this variety? Isn’t there any possibility that it could be a difference between the wines? What methods have you considered to overcome those differences?
Response: We thank the Reviewer for the suggestion. We do apologize for not being able to fully answer to this comment as, unfortunately, it is outside our possibilities to get a complete and conclusive result about the metabolite composition of wines obtained from the very same variety. Too many are the pedoclimatic and technological variables that ultimately affect the metabolite composition of wines, let alone the natural chemical evolution of a single wine over time.
2. Comment: All the representative pictures of western blot analysis should be more clearly demonstrated. The exact methods for the analysis of densitometric intensity should be described in details how arbitrary units (AU) is calculated.
Response: As suggested, the representative pictures of Western blot analysis were improved. As requested, the exact method for the analysis of densitometric intensity was described in the revised manuscript under Material and Methods section, paragraph 4.9, “Cell lysates and Western blot”. See lines 533-537.
3. Comment: In the western blot analysis, all the measurements were done using whole cell lysate? The location of each component looks different. For example, NF-KB showed cytosolic expression, whereas SIRT1 and SIRT6 were expressed mainly in nuclear location. It should better be done using specific section, which the material were mainly expressed.
Response: We thank the Reviewer for the valuable observation and comment. SIRT1 displays nuclear and cytoplasmatic cellular localization and shuttles to the cytoplasm under specific stressful conditions, as well as NF-KB which shows cytosolic expression. SIRT6 is expressed mainly in nuclear location. However, cytoplasmic SIRT6 translocation have been recently provided (So KY, et al Cell Biol Toxicol. 2021 Apr;37(2):193-207. doi: 10.1007/s10565-020-09528-2. Epub 2020 May 11. PMID: 32394328). In light of these observations, we performed two different experimental approaches:
1) Western blot assay using whole cell lysate including both nuclear and cytosolic fractions;
2) Confocal laser scanner microscopy to provide a specific protein (SIRT1, SIRT6, NF-KB, p53) cellular localization.
According to Reviewer suggestion, we highlighted these observations in Results section, paragraph 2.5 “SIRT1 and SIRT6 activation”, and pointed out the attention on sirtuins translocation under Discussion section. See lines 208-212 and 355-356.
4. Comment: In figure 3, mitosox expression in each figure examined by confocal microscopy was not clearly demonstrated. Using arrows might be helpful for clarification. Moreover, detailed method for measurement of specific material expression by confocal laser microscopy representing AU should be described.
Response: As requested, mitosox expression in Figure 3 panel i was clearly defined by using white arrows. Moreover, the method for mitosox expression by confocal laser microscopy was clarified. See Material and Methods section, “Mitochondrial ROS measurement”, paragraph 4.12 and a new Figure 3 j, and Legend to Figure 3 j were provided, and line 203.
5. Comment: SIRT1 is known to be shuttled to the cytoplasm under specific stressful conditions. Wasn’t it the case in this experiment?
Response: In our experiments, under high-glucose and palmitic acid stressful conditions, we observed a decreased nuclear expression of SIRT1, as showed by confocal microscopy assay. According to Reviewer observation, in the revised manuscript we highlighted this observation.
See Results section, paragraph 2.5 “SIRT1 and SIRT6 activation”, lines 208-2012.

Round 2
Reviewer 2 Report
・The figures for TAC and FRAP in Figure 1
・mistake in Figure 2 (b) and (c).
・Choose of Gaglioppo, Magliocco, and Nerello Mascalese
・ antioxidant capacity of Gaglioppo and Nerello Mascalese.
・the target acetylated protein of SIRT1 and SIRT6 .
Most of these points I pointed out last time have been improved, and I think this is a better paper.
Reviewer 3 Report
All the raised issues were responded well.